# Exercise Training Combined with Calanus Oil Supplementation Improves the Central Cardiodynamic Function in Older Women

**DOI:** 10.3390/nu14010149

**Published:** 2021-12-29

**Authors:** Marek Štěpán, Klára Daďová, Miloš Matouš, Eva Krauzová, Lenka Sontáková, Michal Koc, Terje Larsen, Ondrej Kuda, Vladimír Štich, Lenka Rossmeislová, Michaela Šiklová

**Affiliations:** 1Department of Pathophysiology, Centre for Research on Nutrition, Metabolism and Diabetes, Third Faculty of Medicine, Charles University, 10000 Prague, Czech Republic; marek.stepan89@gmail.com (M.Š.); MilosMatous@seznam.cz (M.M.); eva.krauzova@fnkv.cz (E.K.); michal.koc@lf3.cuni.cz (M.K.); vladimir.stich@lf3.cuni.cz (V.Š.); lenka.rossmeislova@lf3.cuni.cz (L.R.); 2Department of Adapted Physical Education and Sports Medicine, Faculty of Physical Education and Sport, Charles University, 16252 Prague, Czech Republic; dadova.klara@gmail.com (K.D.); Sontakova.L@seznam.cz (L.S.); 3Department of Internal Medicine, Královské Vinohrady University Hospital, 10034 Prague, Czech Republic; 4Department of Medical Biology, Faculty of Health Sciences, UiT the Arctic University of Norway, 9037 Tromsø, Norway; terje.larsen@uit.no; 5Institute of Physiology, Czech Academy of Sciences, 14200 Prague, Czech Republic; Ondrej.Kuda@fgu.cas.cz

**Keywords:** aging, omega-3 fatty acids, cardiorespiratory fitness, cardiac output, body composition

## Abstract

The aim of this study was to investigate the possible beneficial effects of exercise training (ET) with omega-3/Calanus oil supplementation on cardiorespiratory and adiposity parameters in elderly women. Fifty-five women (BMI: 19–37 kg/m^2^, 62–80 years old) were recruited and randomly assigned to the 4 month intervention with ET and omega-3 supplementation (Calanus oil, ET-Calanus) or ET and the placebo (sunflower oil; ET-Placebo). The body composition was determined by dual-energy X-ray absorptiometry (DXA), and cardiorespiratory parameters were measured using spiroergometry and PhysioFlow hemodynamic testing. Both interventions resulted in an increased lean mass whereas the fat mass was reduced in the leg and trunk as well as the android and gynoid regions. The content of trunk fat (in percent of the total fat) was lower and the content of the leg fat was higher in the ET-Calanus group compared with the ET-Placebo. Although both interventions resulted in similar improvements in cardiorespiratory fitness (VO_2max_), it was explained by an increased peripheral oxygen extraction (a-vO_2_diff) alone in the ET-Placebo group whereas increased values of both a-vO_2_diff and maximal cardiac output (CO_max_) were observed in the ET-Calanus group. Changes in CO_max_ were associated with changes in systemic vascular resistance, circulating free fatty acids, and the omega-3 index. In conclusion, Calanus oil supplementation during a 4 month ET intervention in elderly women improved the cardiorespiratory function, which was due to combined central and peripheral cardiodynamic mechanisms.

## 1. Introduction

Aging is characterized by a gradual decline in physiological functions. Common changes associated with aging include progressive sarcopenia, a decline in muscle mass and muscular function [1,2], as well as an increase in and redistribution of the fat mass [3]. These changes then contribute to metabolic disturbances, systemic pro-inflammatory states, and an increased risk of metabolic diseases such as type 2 diabetes mellitus (T2DM) and cardiovascular disease (CVD) [3]. Importantly, CVD and associated diseases represent the leading cause of death in people over 65 years [4], and complications associated with CVD significantly impair the quality of life of older people [5].

On the other hand, “active healthy aging” is associated with lower mortality and a higher quality of life for the elderly. An inversely dependent association between muscular strength and physical activity and all-cause mortality in men aged above 60 years shows that those who exercise live longer and have healthier lives [6]. Indeed, physical activity has been shown to improve several aspects of the health of older people including metabolic status [7,8] and cardiorespiratory function [9,10]. Cardiovascular fitness is a strong predictor of CVD [11], and an increase in cardiorespiratory fitness due to physical activity led to a 15–19% reduction of CVD mortality per metabolic equivalent (MET) improvement in “unfit” men [12].

In addition to physical activity, nutrition is another factor that plays an important role in healthy aging. In particular, the effects of omega-3 fatty acids (FA) such as eicosapentaenoic acid (EPA) and docosahexaenoic acid (DHA) on various age-associated changes of health have been widely studied in the last two decades. Although the outcomes of epidemiological studies are not consistent, showing either positive or no effects of omega-3 FA on CVD risk of death [13,14], the beneficial effects of omega-3 FA on cardiac function during exercise (e.g., stroke volume, cardiac output) have been demonstrated in healthy individuals [15]. The effects of DHA-enriched food supplementation on the antioxidative capacity and oxidative damage have been observed in football players during a training season [16]. In mice, omega-3 FA modulated inflammation and the FA composition in the heart, which alleviated cardiac dysfunction and fibrosis [17]. Calanus oil is a relatively new dietary supplement of omega-3 FA, and its positive anti-inflammatory and cardiometabolic effects have been reported in mice [18]. Furthermore, Calanus oil restored cardiac metabolic flexibility and improved the recovery of the cardiac function following an ischemia in mice on a high-fat diet [19]. Calanus oil is extracted from planktonic *Calanus finmarchius* [20], representing probably the largest source of omega-3 FA on the planet. It has a unique chemical composition (Table 1). The content of EPA (20:5 n-3) and DHA (22:6 n-3) is relatively low compared with other marine oils but it has a high content of stearidonic acid (SDA, 18:4 n-3) as well as other long-chain monounsaturated fatty acids such as gondoic acid (20:1 n-9) and cetoleic acid (22:1 n-11) [21]. Furthermore, the fatty acids in Calanus oil are mostly bound as monoesters (wax esters) where the fatty acids are linked to the monounsaturated long-chain fatty alcohols eicosenol (20:1 n-9) and docosenol (22:1 n-11) [22]. In krill and fish oil, the larger part of the fatty acids is bound in phospholipids and triglycerides, respectively. Calanus oil also contains proteins, vitamins, minerals, and phytosterols as well as a high amount of the antioxidant astaxanthin [23]; the latter gives the oil its characteristic red color.

Combining exercise training (ET) with Calanus oil supplementation could represent a better strategy for improving the cardiorespiratory function in older people than ET alone. Therefore, in this study, we aimed to investigate the potential beneficial effects of combined aerobic/resistance ET with Calanus oil supplementation on cardiorespiratory parameters and body composition in older women when compared with the effects of ET with a placebo supplementation.

## 2. Materials and Methods

### 2.1. Subjects and Study Design

The study was part of the EXODYA (Effect of Exercise Training and Omega-3 Fatty Acids on Metabolic Health and Dysfunction of Adipose Tissue in the Elderly) research project, Clinical Registration No: NCT03386461. A total of 213 women initially responded to the advertisement and required detailed information about the study. Of the 213 women, 127 were rejected either based on the exclusion criteria or because they lost interest in participating upon reviewing the study protocol. For the intervention, 55 elderly women (aged 62–80 years) were randomly assigned into 2 groups with the same baseline anthropometric characteristics. The exclusion criteria were: weight change more than 3 kg within 3 months preceding the study; smoking; diagnosed cancer, diabetes, liver, and renal diseases; untreated hyper- or hypothyroidism; steroid use; use of beta-blockers; use of omega-3 dietary supplements; and long term use of anti-inflammatory medication (anti-rheumatics and analgesics affecting cyclooxygenases; 100 mg of anopyrin daily was accepted). Subjects taking medication to lower cholesterol levels and blood pressure (representing 70–90% of the older population) were permitted. The scheme of the study design was recently published [24].

The participants engaged in a 4 month exercise program. The training included supervised 60 min lessons of circuit training, postural exercises, and weight-bearing exercises in a gym twice a week as well as an aerobic session of Nordic walking once a week. A detailed description of the training was reported before [25]. In addition to the training, the protocol group 1 (ET-Calanus) was provided with omega-3 wax ester-rich supplementation in the form of Calanus oil (5 capsules/day of Calanus oil, Calanus AS, Norway) and group 2 (ET-Placebo) was supplemented with placebo capsules (5 capsules/day of pure sunflower oil, Ayanda GMBH & CO. KG, Pritzwalk, Germany). The dose of supplementation was chosen to achieve a good adherence by the participants and to provide a daily recommended dose for preventive effects on cardiovascular health (approximately 230 mg/day of EPA + DHA) [26]. The fatty acid composition of the capsules is listed in Table 1.

At the baseline, the groups were not different with respect to the nutritional and physical activity status (Table 2). The dietary intake of the participants was monitored by three-day dietary records before and at the end of the study. The omega-3 FA consumption was analyzed by a questionnaire that focused on omega-3-rich foods and by the measurement of the omega-3 index [27]. The dietary habits were not different between the groups at the baseline and did not change during the intervention (Table 2). The physical activity level was assessed by senior fitness tests and an exercise stress test [25] (Table 2). Adherence to the capsule consumption was monitored by self-reporting and the lipidomic analysis of the plasma [28].

A total of 53 women completed the intervention; the drop-outs (*n* = 2) were caused by newly diagnosed health problems that were not associated with the ET itself. Two women did not undergo all clinical examinations at the end of the intervention for personal reasons; therefore, data from 51 women were finally analyzed.

### 2.2. Clinical Examinations

The subjects were examined at 8 am after an overnight fast at the baseline (before the start of the training) and after 4 months of the intervention. The clinical examination and collection of blood samples were performed at least 48 h after the previous exercise session and physical performance testing. After 15 min of rest, blood was drawn from the antecubital vein. Sera were obtained after 30 min of clotting followed by centrifugation at 1000× *g* for 10 min. The body composition was assessed by DXA technology using a Lunar iDXA (GE Healthcare, Madison, WI, USA). All selected data were analyzed with enCORE software (GE Healthcare, Madison, WI, USA). For the regional fat analysis, the regional boundaries were defined similarly as described by Stults-Kolehmainen et al. [29]. Briefly, the arm region included the arm and shoulder area. The trunk region included the neck and chest as well as the abdominal and pelvic areas. The leg region included the whole area below the pelvic area. The android region was the area between the ribs and the pelvis and was a part of the trunk region. The gynoid region included the hips and upper thighs, and overlapped both the leg and trunk regions.

### 2.3. Bicycle Ergometry with PhysioFlow Hemodynamic Testing

Resting blood pressure and resting electrocardiography (ECG; Schiller, Baar, Switzerland) were obtained prior to spiroergometry testing. Each subject performed an exercise test on the bicycle ergometer Ergoselect 200 (Ergoline, Bitz, Germany), as described before [25]. Throughout the test, the heart rate (HR), blood pressure, and ECG were recorded. Oxygen consumption, carbon dioxide output, and expired ventilation (VE) were measured by a breath-by-breath gas exchange analyzer PowerCube-Ergo (Ganshorn Medizine Electronic GmbH, Niederlauer, Germany). The indices of the myocardial functions were evaluated by impedance cardiography (PhysioFlow, Manatec Biomedical, Paris, France). The analysis ran in association with an ECG signal and, therefore, it related the information to the cardiac cycle and function [30,31,32]. The stroke volume (SV), cardiac output (CO), and systemic vascular resistance (SVR) were analyzed. The peripheral oxygen extraction (a-vO_2diff_) was calculated according to Fick’s principle as a-vO_2diff_ (mL O_2_/100 mL blood) = VO_2max_ (l/min)/COmax (l/min) × 100. Due to problems with the skin electrode connection in several participants, a few bioimpedance measurements from PhysioFlow did not meet the criteria for sufficient signal validity and these data were not analyzed. Thus, the cardiorespiratory parameters were reported in 16–24 patients in the ET-Placebo group and 24–25 patients in the ET-Calanus group. 

### 2.4. Plasma Analysis

The serum concentration of leptin was determined using an ELISA kit (Duoset Leptin, R&D Systems, Minneapolis, MN, USA). The coefficient of the variation of the assay was below 2%. Free fatty acid (FFA) levels were measured using enzymatic colorimetric kits (Randox, Crumlin, United Kingdom). The analysis of the lipidomic profile and omega-3 index was performed as described before [28].

### 2.5. Statistics

The data were presented as mean ± SD or SEM. The statistical analysis was performed using GraphPad Prism 9.1.2 for Windows (La Jolla, CA, USA). The differences between the responses to interventions in the 2 groups of subjects (ET-Placebo vs. ET-Calanus) were analyzed by a repeated measure two-way analysis of variance (ANOVA) with a Bonferroni post-hoc analysis; the data were log-transformed for normalization. A comparison of the percent changes between the groups (ET-Placebo vs. ET-Calanus) was performed by an unpaired Mann–Whitney rank test and a multiple Mann–Whitney test. The correlations were expressed as a Pearson’s correlation coefficient. The level of significance was set at *p* < 0.05.

## 3. Results

### 3.1. Exercise Training Combined with Calanus Oil Supplementation Affects the Body Composition

First, we examined the effects of ET intervention with Calanus oil or placebo supplementation on the body composition parameters. We also evaluated the levels of FFA and adipokine leptin, which are related to the adipose tissue mass and metabolism.

At the baseline, the groups did not differ in nutritional and anthropometric parameters, as shown in Table 2 and Table 3 and reported recently [24,25]. A significant increase in EPA and lipids containing EPA and DHA was present in the Calanus group when compared with the placebo group (Appendix A). In response to the interventions, both groups showed a decreased total body weight and fat mass. Similarly, the plasma levels of leptin decreased during the intervention resulting in a positive correlation between the relative change in leptin and the relative change in the fat mass (r = 0.518; *p* = 0.004).

Furthermore, a small increase in the lean mass was observed in response to the intervention in both groups with a post-hoc significant effect in the ET-Calanus group (Table 3).

An interaction between ET and supplementation was observed for the ratio of the trunk fat and leg fat related to the total fat mass, which indicated a change in the fat distribution in the ET-Calanus group (a decrease in the percentage of the trunk fat and an increase in the percentage of the leg fat related to the total fat). A trend toward a greater reduction in the ratio of android fat to the total fat was observed in the ET-Calanus group compared with the ET-Placebo group (time*group interaction: *p* = 0.066) (Table 3).

### 3.2. Exercise Training Combined with Calanus Supplementation Affects the Central Cardiorespiratory Function

We then examined the effects of combined aerobic and resistance ET with Calanus oil or the placebo on cardiorespiratory fitness and the relative contribution of central mechanisms; i.e., cardiac output (CO) and the peripheral arterial–venous O_2_ difference (a-vO_2diff_) affecting this parameter.

Cardiorespiratory fitness, expressed as VO_2max_, as well as ventilation increased in both groups irrespective of the type of supplementation (Table 4, Figure 1). An increase in a-vO_2diff_ was present in both groups whereas CO_max_—and, hence, stroke volume (SV_max_) —increased only in the ET-Calanus group (Table 4, Figure 1). An improvement in all these parameters was not associated with the age of the participants (data not shown).

When looking to the predictors of changes in CO_max,_ in response to the intervention, we performed correlations and a multiple regression analysis. The multiple regression showed that SVR and circulating FFA levels significantly predicted a change in CO_max_ (β = −0.41, *p* = < 0.001 and β = 0.26, *p* = < 0.001, respectively) whereas age and body composition were not significantly associated with a CO_max_ change (data not shown). The relative changes in CO_max_ correlated positively with changes in the omega-3 index (Figure 2a). Although the changes in CO_max_ were not correlated with the changes in leptin concentration, the plasma levels of leptin correlated with CO_max_ before and at the end of the interventions (Figure 2b).

## 4. Discussion

The main finding of this study was that supplementation with Calanus oil altered the mechanism of the exercise-induced increase in cardiorespiratory fitness in elderly women. Although ET with Calanus oil administration affected both the central (CO_max_) and peripheral mechanisms (a-vO2_diff_), ET with the placebo supplementation only affected the peripheral oxygen extraction. Apart from these findings, we have shown an additional effect of ET when combined with Calanus oil supplementation on several parameters of body composition such as lean body mass and fat distribution in the trunk and leg region.

Our findings showed that the addition of Calanus oil supplementation enforced the improvement of the cardiac output (CO_max_) without a reinforcing effect on VO_2max_. The increased CO_max_ could be explained in terms of an increased SV (probably also a reduced SVR). The central and peripheral mechanisms causing improved cardiovascular fitness in response to ET have been reported to vary between genders as well as age groups [33,34] but in the current study, the potential to improve the cardiorespiratory parameters was not dependent on the age of the women. This was in line with the results of Støren et al. [35] who suggested that people across all age groups have the same potential for cardiovascular improvement.

One of the important contributors to the improvement in CO_max_ in the ET-Calanus group might be the use of Calanus oil itself with its unique mixture of fatty acids bound to fatty alcohols in the chemical form of wax esters [20,22]. The molecular effects of Calanus compounds are not well-explored; however, its beneficial effects on inflammation and the cardiovascular system have been shown previously in mice [18,19,36]. Notably, two important components of Calanus oil, eicosapentaenoic acid (EPA) and docosahexaenoic acid (DHA), have known effects on inflammation and oxidative stress [37,38], which are thought to underlie the pathogenesis of cardiovascular disease through the induction of DNA damage and protein fragmentation and dysfunction [39]. These events contribute to left ventricular hypertrophy and remodeling that exacerbate cardiac dysfunction. Indeed, in the study of Walser et al., supplementation with EPA and DHA for 6 weeks increased SV_max_ and CO_max_ in response to a low or moderate exercise load in healthy adults [15].

Altered levels of substrates and circulating hormones could probably contribute to an improvement in myocardial function and maintaining the ability of the heart to keep up with increasing and repeated workloads [40]. The association between changes in CO_max_ and plasma FFA as well as the baseline and post-ET correlation between CO_max_ and leptin supports such a molecular mechanism and thereby a possible crosstalk with adipose tissue, which is the main source of these factors [41]. Indeed, fatty acid metabolism may have contributed to the increased CO_max_ in response to the intervention. In a healthy myocardium, fatty acid oxidation provides 50–70% of the cardiac energy requirement and metabolic adaptations to ET include elevated fatty acid oxidation [42]. Thus, the positive correlation between the plasma FFA and CO_max_ during ET combined with Calanus oil might have supported the currently observed improvement in the cardiac function. Similarly, leptin has been reported to have beneficial effects on glucose and fatty acid metabolism in the heart [43] and to protect against triglyceride accumulation and lipotoxicity [44].

We also observed a change in body phenotype toward a lower trunk fat mass and higher lean mass in the ET-Calanus group. A similar effect of Calanus oil combined with ET was recently reported by Wasserfurth et al. [45]. In line with this, we also showed a stronger effect of the combined ET-Calanus intervention on leg muscle strength in a previous publication [25]. These findings together suggest a higher muscular anabolic potential and strength/function of exercise with a concomitant use of Calanus oil. The fact that omega-3 FA potentiates muscle turnover is not new [46]. There are studies bringing proof that omega-3 supplementation attenuates muscle tissue atrophy and can promote muscle protein synthesis through the incorporation of EPA and DHA into phospholipid membranes and intracellular organelles, resulting in an improved muscle protein synthesis [46]. Moreover, a significant reduction in the trunk fat and possibly android fat might be important from the view that the fat content in this region is directly associated with the visceral fat content and concomitantly the risk of metabolic diseases [47]. Thus, the molecular mechanism of the effect of Calanus oil on visceral fat accumulation as well as muscle mass and function should be evaluated in future studies.

There are several limitations to the study. One of them is that a control non-exercising group was missing. However, for ethical and feasibility reasons, we decided to compare the current interventions to bring a greater benefit to the elderly subjects than omega-3 supplementation alone. Moreover, PhysioFlow, i.e., a bioimpedance based method, is not considered to be the “gold standard” for the measurement of cardiovascular function during an acute bout of exercise. However, the method was shown to be a reliable alternative to other methods demonstrated before [30,31,32]. Importantly, the present study included only women but it confirmed the results from several previous exercise intervention studies [34,48,49].

## 5. Conclusions

In conclusion, the importance behind this study is that Calanus oil supplementation combined with exercise training over just four months improved the response of the central cardiodynamic function to a maximal workload, specifically the maximal cardiac output, whereas ET with a placebo supplementation worked primarily through a peripheral oxygen extraction mechanism. Moreover, the beneficial effects of ET and Calanus oil supplementation on body composition were confirmed. These changes have a significant clinical potential to improve the quality of life of older people and potentially reduce CVD risk and mortality.

## Figures and Tables

**Figure 1 nutrients-14-00149-f001:**
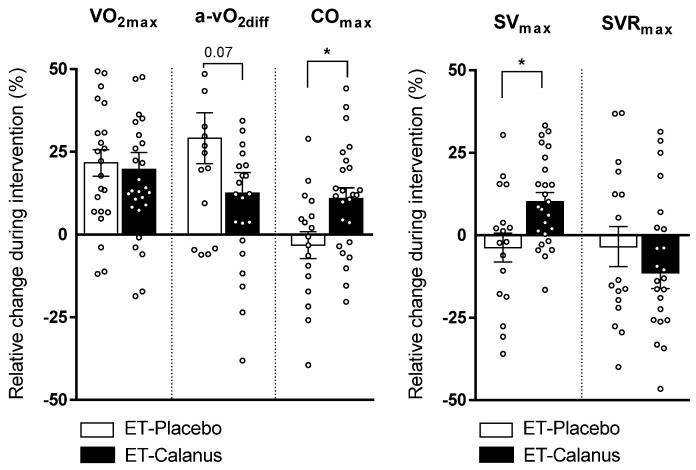
Individual relative changes in cardiorespiratory parameters during the intervention in ET-Placebo (*n* = 16–24) and ET-Calanus groups (*n* = 24–27). Data are presented as mean ± SEM; * *p* < 0.05 (Mann–Whitney test). VO_2max_: maximal oxygen consumption; a-vO_2diff_: peripheral extraction of oxygen; CO_max_: maximal cardiac output; SV_max_: maximal stroke volume; SVR_max_: maximal systemic vascular resistance.

**Figure 2 nutrients-14-00149-f002:**
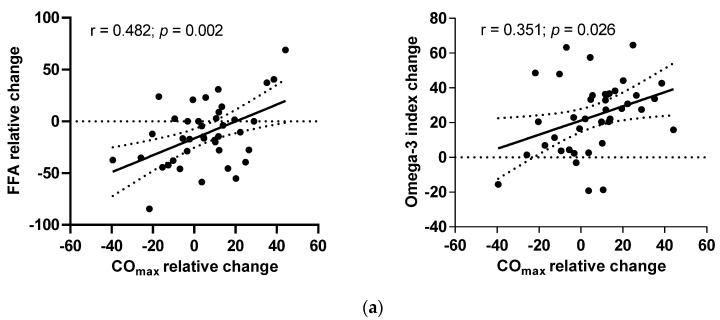
Correlations of (**a**) relative changes in maximal cardiac output (CO_max_) with relative changes in free fatty acids (FFA) in ET-Placebo and ET-Calanus groups during the intervention; (**b**) CO_max_ with plasma leptin levels before and after the intervention in all subjects. The Pearson’s correlation coefficient is presented (ET-Placebo, *n* = 16; ET-Calanus group, *n* = 24).

**Table 1 nutrients-14-00149-t001:** Fatty acid composition of Calanus Oil and sunflower oil in the capsules.

Fatty Acid		Calanus Oil	Sunflower Oil
		% of Total Fatty Acids	
Myristic acid	14:0	15.37	nd
Palmitic acid	16:0	12.84	6.3
Stearic acid	18:0	0.85	4.6
Arachidic acid	20:0	0.10	0.5
Palmitoleic acid	16:1 n-7	3.80	0.3
Oleic acid	18:1 n-9	4.32	26.7
Gondoic acid	20:1 n-9	3.63	nd
Gadoleic acid	20:1 n-11	3.90	nd
Cetoleic acid	22:1 n-11	5.06	nd
Nervonic acid	24:1 n-9	0.58	nd
Linoleic acid	18:2 n-6	1.65	61.1
Linolenic acid	18:3 n-3	4.38	0.3
Stearidonic acid	18:4 n-3	20.50	nd
Arachidonic acid	20:4 n-3	0.38	nd
Eicosapentaenic acid	20:5 n-3	10.98	nd
Docosahexaenoic acid	22:6 n-3	9.26	nd
*Sum of fatty acids*			
SFA (g/100g oil)		14.7	12.3
MUFA (g/100g oil)		14.0	28.7
PUFA (g/100g oil)		20.4	59.0
Fatty alcohol		39.0	nd

nd: not detected.

**Table 2 nutrients-14-00149-t002:** Nutritional and physical activity characteristics of the subjects before and at the end of the study.

	ET-Placebo		ET-Calanus		Two-Way ANOVA *p*-Value		
	Before	After	Before	After	Time	Group	Interaction
Age	70 ± 4		71 ± 4			0.33	
Nutrition							
Energy (kJ/day)	7201 ± 1892	7385 ± 1276	7716 ± 991	7531 ± 1262	0.78	0.23	0.13
Carbohydrates (g)	205 ± 38	210 ± 43	223 ± 37	218 ± 39	0.98	0.15	0.34
Fat (g)	69 ± 33	71 ± 22	72 ± 11	74 ± 15	0.20	0.19	0.34
Protein (g)	72 ± 25	73 ± 15	78 ± 12	72 ± 15	0.67	0.35	0.06
Physical activity							
Chair-stand test (repetitions)	16 ± 4	19 ± 4	16 ± 4	20 ± 5	<0.001	0.99	0.03
Arm-curl test (repetitions)	19 ± 4	24 ± 4	19 ± 4	24 ± 5	<0.001	0.58	0.46

Data are presented as mean ± SD. Statistical differences of log-transformed data were evaluated by RM two-way ANOVA with a Bonferroni post-hoc analysis; *p*-value of the statistical difference in the main effect (time, group, time × group interaction).

**Table 3 nutrients-14-00149-t003:** Anthropometric characteristics, fat distribution, and circulating factors during the interventions.

Parameter	ET-Placebo (*n* = 23–24)		ET-Calanus (*n* = 27)		Two-Way ANOVA *p*-Value		
	Before	After	Before	After	Time Effect	Group/Suppl. Effect	Interaction
Fat mass (%)	38.8 ± 5.2	37.4 ± 5.1 ***	41.4 ± 5.1	40.2 ± 5.7 ***	<0.001	0.11	0.84
Total body weight (kg)	71.6 ± 12.9	70.5 ± 12.7 **	71.5 ± 10.1	71.0 ± 10.6	0.003	0.70	0.16
Fat mass (kg)	27.6 ± 8.3	26.0 ± 7.9 ***	29.3 ± 7.7	28.2 ± 7.7 ***	<0.001	0.31	0.70
Lean mass (kg)	42.0 ± 5.9	42.3 ± 5.7	40.4 ± 3.4	40.9 ± 3.1 *	0.002	0.35	0.72
Arm fat (kg)	3.00 ± 0.99	2.89 ± 0.81	3.13 ± 0.66	3.08 ± 0.79	0.15	0.34	0.97
Leg fat (kg)	9.23 ± 3.06	8.72 ± 2.92 ***	9.86 ± 2.83	9.68 ± 2.71	<0.001	0.23	0.05
Trunk fat (kg)	14.3 ± 4.9	13.6 ± 4.6 *	15.4 ± 5.4	14.6 ± 5.2 **	<0.001	0.43	0.61
Android fat (kg)	2.36 ± 0.96	2.22 ± 0.89	2.49 ± 1.00	2.37 ± 1.02 *	<0.001	0.50	0.43
Gynoid fat (kg)	4.36 ± 1.23	4.14 ± 1.20 **	4.77 ± 1.12	4.56 ± 1.23 **	<0.001	0.19	0.83
Arm fat/total FM (%)	10.9 ± 1.5	11.2 ± 1.5	10.9 ± 1.7	11.1 ± 1.5	0.25	0.77	0.85
Leg fat/total FM (%)	33.9 ± 5.6	33.6 ± 5.4	34.0 ± 5.8	35.0 ± 6.2 *	0.14	0.69	0.02
Trunk fat/total FM (%)	51.6 ± 6.1	51.6 ± 5.9	52.0 ± 5.8	50.6 ± 6.4 **	0.04	0.86	0.03
Android fat/total FM (%)	8.2 ± 1.6	8.3 ± 1.6	8.3 ± 1.5	8.1 ± 1.6	0.27	0.99	0.07
Gynoid fat/total FM (%)	16.1 ± 2.3	16.0 ± 2.0	16.4 ± 2.1	16.3 ± 2.0	0.48	0.60	0.83
Plasma FFA (mmol/L)	0.65 ± 0.24	0.51 ± 0.21 *	0.65 ± 0.24	0.61 ± 0.21	0.006	0.12	0.47
Plasma Leptin (ng/mL)	20.3 ± 12.0	16.0 ± 10.0 *	21.1 ± 12.0	19.5 ± 11.3	0.015	0.58	0.39
Omega-3 index (%)	5.22 ± 1.25	6.10 ± 1.66	4.99 ± 0.98	6.39 ± 1.17	<0.001	0.787	0.06

Data are presented as mean ± SD. Statistical differences of log-transformed data were evaluated by RM two-way ANOVA with a Bonferroni post-hoc analysis; *p*-value of the statistical difference in the main effect (time, group, time × group interaction). *** *p* < 0.001, ** *p* < 0.01, * *p* < 0.05 post-hoc statistically significant difference during the intervention (before vs. after). FM: fat mass; ET: exercise training; FFA: free fatty acids. Omega-3 index: content of omega-3 FA in erythrocyte membranes in percent of the total FA.

**Table 4 nutrients-14-00149-t004:** Cardiovascular and ventilation parameters of participants before and after the intervention.

Parameter	ET-Placebo (*n* = 16–24)		ET-Calanus (*n* = 24–27)		Two-Way ANOVA		
	Before	After	Before	After	Time Effect	Group Effect	Interaction
VO_2max_ (mL/kg/min)	19.3 ± 3.1	22.9 ± 3.1 ***	19.6 ± 4.2	23.0 ± 4.6 ***	<0.001	0.92	0.71
VE_max_ (L/min)	56.8 ± 14.1	65.7 ± 14.3 ***	56.0 ± 8.5	70.3 ± 12.1 ***	<0.001	0.45	0.17
HR_rest_ (bpm)	70.5 ± 7.1	69.0 ± 7.7	72.3 ± 11.9	71.0 ± 10.4	0.37	0.52	0.86
HR_max_ (bpm)	156.1 ± 12.9	154.8 ± 11.5	152.2 ± 14.3	156.3 ± 13.1 **	0.14	0.72	0.011
SV_rest_ (mL per beat)	75.3 ± 12.4	72.6 ± 12.3	75.2 ± 16.2	77.2 ± 14.2	0.87	0.61	0.16
SV_max_ (mL per beat)	104.0 ± 22.4	97.8 ± 16.0	106.1 ± 21.9	114.9 ± 16.5 *	0.49	0.08	0.005
CO_rest_ (L/min)	6.03 ± 1.01	5.63 ± 1.07	6.34 ± 1.60	6.17 ± 1.42	0.09	0.32	0.35
CO_max_ (L/min)	16.0 ± 3.6	15.0 ± 2.9	15.7 ± 3.5	17.0 ± 2.7 *	0.59	0.33	0.006
Maximal a-vO_2_ difference (mL O_2_/100 mL blood)	9.0 ± 1.7	11.5 ± 2.7 ***	8.9 ± 2.4	9.7 ± 2.4	<0.001	0.17	0.049
SVR_rest_ (dys.s/cm^2^)	1247 ± 238	1210 ± 238	1196 ± 216	1180 ± 243	0.45	0.58	0.82
SVR_max_ (dys.s/cm^2^)	492 ± 93	489 ± 94	495 ± 102	455 ± 102	0.23	0.51	0.29

Data are presented as mean ± SD. Statistical differences of log-transformed data were evaluated by RM two-way ANOVA with a Bonferroni post-hoc analysis; *p*-value of statistical difference in the main effect (time, group, time × group interaction). *** *p* < 0.001, ** *p* < 0.01, * *p* < 0.05 post-hoc statistically significant difference during the intervention (before vs. after). ET: exercise training; VO_2max_: maximal oxygen consumption; V_Emax_: maximal minute ventilation, HR_rest_: resting heart rate; HR_max_: maximal heart rate; SV_rest_: resting stroke volume; SV_max_: maximal stroke volume; CO_rest_: resting cardiac output; CO_max_: maximal cardiac output; a-vO_2_: arteriovenous oxygen difference; SVR_rest_: resting systemic vascular resistance; SVR_max_: maximal systemic vascular resistance.

## Data Availability

Data generated or analyzed during this study are available from the corresponding author upon reasonable request.

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
