# Peer review of "Exercise Training Combined with Calanus Oil Supplementation Improves the Central Cardiodynamic Function in Older Women"

_nutrients, 2021, doi:10.3390/nu14010149_

Round 1

Reviewer 1 Report

Authors examined effects of calaus oil supplementation when it is used with exercise training in a randomized controlled trial in older women.

Line 80-82, “55 elderly women (aged 62- 80 years), were randomly assigned into 2 groups with the same baseline anthropometric characteristics.”

The intervention includes both nutritional and physical training. The two groups should be similar in nutritional and physical activity condition; ie., if some people already had consumed enough amount of omega-3 fatty acids before the study and were assigned more to either of the groups, the observed effect of calanus oil supplementation might have been smaller. Data on nutritional intake and physical activity level before the intervention for both groups should be presented.

Line 93-97, “group 1 (ET-Calanus) was provided with omega-3 wax esters rich supplementation in the form of Calanus oil (5 capsules/day of Calanus oil, Calanus a.s., Norway) and group 2 (ET-Placebo) was supplemented by “placebo” capsules (5 capsules/day of pure Sunflower oil, Calanus a.s., Norway)”

Basic information about the ingredients of the capsules should be described; ie., total amount of fatty acids, composition of the fatty acids in the each capsules.

Line 145-146, The level of significance was set at p < 0.05.

P values for leg fat and android fat / total FM are not significant. P values should be presented for all variables, not “NS”, in the Tables. The presentations seem arbitrary.

Reviewer 2 Report

The authors have provided little information on the source or the fatty acid content of the novel "calanus oil", it may be published elsewhere but it is directly relevant to this paper and should be included. As you have not described the calanus oil, and how it differs to fish oil, there is little rational as to why this study should be done. While the data on fatty acid levels of calanus oil can be provided, there are serious flaws with the design of the study which can not be rectified:

  1. They have not indicated whether any differences existed in the background diets of the two groups, especially in relation to the n-3 fatty acid content.
  2. They have not monitored the adherence/compliance to taking the supplements.
  3. They have not demonstrated whether the changes in cardiorespiratory function are in relation to any changes seen in tissue levels of n-3 fatty acids because of the consumption of the calnus oil. Measuring the total plasma FFA does not tell us about the changes in n-3 fatty acids.
  4. When discussing n-3 fatty acids you need to be able to differentiate which ones you are referring to in your studies and also from the evidence base(introduction), is it ALA, EPA, DHA or DPA.

Round 2

Reviewer 1 Report

The manuscript has been revised and improved sufficiently.

Author Response

We thank to reviewer for accepting of our corrections and for possibility to improve the manuscript.

Reviewer 2 Report

Arachidonic acid levels are not presnet in the oils (table 1)

Need evidence that there was no difference in dietary intakes during the study. 

Omega- 3 levels need to be expressed as either g/day or % of E. Please specify.

You have only presented total FFA values - we need the distribution of all the fatty acids in the plasma - so how much, including but not limted to LA, ALA, SA, OA, AA, EPA, DHA, DPA etc.... This will let us know what changes have occurred in relation to the consumption of calanus oil, otherwise it is mere specualtion that the omega 3 influenced cardiodynamic function.
